# Patient satisfaction with healthcare services among health insurance program beneficiaries in Nepal: A cross-sectional study

Biraj Man Karmacharya[1,2]*, Sabina Marasini[1,2]*, Ruby Maka Shrestha[1], Sudim Sharma[3], Mukesh Adhikari[4], Samip Pandey[1], Sambhu Prasad Jnawali[5], Deepa Guragain[1], Ishwori Byanju Shrestha[1], Anjali Joshi[1,2], Bihari Sharan Kuikel[1,2], Nripa Raj Dangaura[1,2], Madan Kumar Upadhyaya[6], Upama Baral[5], Pramesh Koju[1], Dong Roman Xu[6,7,8,9]

1 Department of Public Health and Community Programs, Dhulikhel Hospital, Dhulikhel, Nepal, 2 Kathmandu University School of Medical Sciences, Dhulikhel, Nepal, 3 Group for Technical Assistance, Lalitpur, Nepal, 4 Department of Health Policy and Management, Gillings School of Public Health, University of North Carolina, Chapel Hill, North Carolina, United States of America, 5 Health Insurance Board, Kathmandu, Nepal, 6 Ministry of Health and Population, Nepal, 7 Acacia Lab for Implementation Science, School of Health Management and Dermatology Hospital, Southern Medical University, Guangzhou, China, 8 Center for World Health Organization Studies and Department of Health Management, School of Health Management of Southern Medical University, Guangzhou, China, 9 Southern Medical University Institute for Global Health (SIGHT), Dermatology Hospital of Southern Medical University (SMU), Guangzhou, China

* birajmk@kusms.edu.np (BMK); shabinam367@gmail.com (SM)

## Abstract

### Background

Patient satisfaction, often assessed through clients' experiences and opinions are vital for improving healthcare services, shaping health policies, and providing feedback on the quality, availability, and responsiveness of healthcare services. In this study, we assessed the healthcare satisfaction levels of insured patients with the health services provided under the National Health Insurance Program (NHIP) accredited health facilities.

### Methods

We conducted a cross sectional study at 22 health facilities across 3 provinces of Nepal. We utilized client-exit interviews among 468 patients enrolled and utilizing the health services under NHIP. We used a validated Patient Satisfaction Questionnaire III (PSQ-III) developed by RAND Corporation including various contextual socio-demographic characteristics. We calculated mean score and percentages of satisfaction across seven dimensions of patient satisfaction. To determine the association between various dimensions of patient satisfaction and socio- demographic characteristics of the patients, we used generalized ordered logistic regressions.

**Data availability statement:** All relevant data are within the manuscript and its Supporting Information files.

**Funding:** The study was conducted with the funding support from Swiss Agency for Development and Cooperation (SDC). The funders had no role in study design, data collection and analysis, decision to publish, or preparation of the manuscript.

**Competing interests:** The authors have declared that no competing interests exist.

## Results

Among 468 patients, we observed a wide variation in patient satisfaction across seven dimensions. About 87% of the patients were satisfied in the domain of inter-personal manner, 83% in technical quality, 63% in accessibility and convenience and 60% in financial aspects. The overall satisfaction was observed as 74%. The patients having chronic diseases among the family members were more satisfied compared to those having no chronic disease in the family members (AOR:5.96; 95% CI: 2.65–13.39). Presence of chronic disease and subsidy status were found to be associated with patient satisfaction in most dimensions.

## Conclusions

Patient satisfaction measures the gap between expected and actual service experience, tied to service quality, availability, accessibility, and financial aspects. While NHIP aims to provide quality services, the actual service quality mainly depends on the efforts of the health facilities. Therefore, strengthening the health system is crucial for improving service quality and ensuring user retention and satisfaction with NHIP.

## Introduction

Patient satisfaction is a multidimensional concept that encompasses individuals' perceptions, expectations, and experiences. It is an important component of health-care quality reflecting the ability of healthcare providers to meet patient's needs and expectations. In many countries assessment and measurement of patient satisfaction with the health care system is recognized as the key indicator of health care quality [1]. Measuring healthcare quality through patient satisfaction helps to improve clinical outcomes, patient retention, and malpractice claims. It also guides health-care delivery system to be more timely, efficient, and patient-centered [2]. Improving patient satisfaction aims to achieve three main goals: enhancing the patient experience; improving population health; and reducing per capita healthcare costs [3]. In many countries, out-of-pocket expenses continue to be a significant barrier to accessing healthcare, causing financial strain and inequitable health outcomes. To address these challenges, governments have implemented health insurance scheme designed to provide financial protection and improve access to healthcare services.

The Government of Nepal endorsed the National Health Insurance Policy in 2014, and in 2016, launched the Social Health Insurance program, now known as the National Health Insurance Program (NHIP) [4]. Guided by the Health Insurance Act of 2017 and its regulations in 2019, NHIP is administered by the Health Insurance Board (HIB), which oversees the provision of services through a network of 440 accredited health facilities, both public and private, across the country. This program aims to improve healthcare service quality, broaden access to essential health services, and alleviate the financial burden on households, particularly given the significant 54.2% out-of-pocket payments within current health expenditure [5]. Despite the

NHIP's target of 60% population enrollment by 2022, only 21.4% enrollment had been achieved by mid-2022, with annual dropout rates at around 25% [6]. Factors contributing to poor enrollment and high dropout rates include the unavailability of drugs, poor service quality, unfriendly behavior of health workers, and indifferent treatment of insured patients in public health facilities [7].

To address these challenges and ensure the sustainability of the insurance program, improving service quality is as crucial as expanding insurance coverage. Service providers and health insurance board must systematically measure user satisfaction to evaluate their services' effectiveness because clients' experiences and opinions are vital for improving healthcare services, shaping health policies, and providing feedback on the quality, availability, and responsiveness of healthcare services [8]. Recognizing its benefits, developed countries widely use patient satisfaction as an index of health-care quality [9]. In contrast, low-and-middle income countries (LMICs) have limited use of patient satisfaction metrics. Due to resource constraints, these countries focus more on basic supplies and services than on quality. Similar to other LMICs, Nepal also has not prioritized improving patient satisfaction. A survey of health facilities in Nepal found that only 23.2% of health facilities tend to have quality assurance activities and only 4% have a functional client feedback system [10]. Given Nepal's unique socio-economic and health system context, coupled with its incipient national health insurance program, the current evidence base for patient satisfaction towards health system and the NHIP, in particular, remains limited. Most studies related to NHIP have focused on determinants of enrollment, dropout rates, service utilization but very few have examined the patient satisfaction [4,7,11–13]. Therefore, this study seeks to assess the healthcare satisfaction levels of insured patients with the health services provided under the NHIP accredited health facilities. By examining patient experiences and satisfaction, the study aims to provide valuable insights into the program's strengths and weaknesses, informing policymakers, service providers and stakeholders in their efforts to enhance the quality and accessibility of healthcare services for all citizens. The findings will contribute to the ongoing discourse on health insurance effectiveness and the broader goal of achieving Universal Health Coverage (UHC) in Nepal.

## Materials and methods

### Study design and settings

We conducted a cross-sectional study in 22 health facilities (S1 Table) across three provinces (Madhesh, Bagmati and Karnali) of Nepal. For the selection of provinces, we used the health insurance enrollment percentage and health service utilization data from all seven provinces, as reported in the Health Insurance Board's Annual Report 2021/22 [6]. In consultation with the authorities at the HIB, we selected three provinces that had high, moderate and low rates of health insurance enrollment and health service utilization. Within the provinces, we selected three types of health facilities: primary healthcare centres (PHCC), private/community/teaching hospitals and public hospitals. These are the primary service points determined by the NHIP. The selection of health facilities was also done based on the patient flow and reimbursement claims (high and low) of the health facilities, in consultation with the HIB.

### Sample size and recruitment

We conducted client-exit interviews with the patients enrolled under the NHIP scheme. A total of 468 insured patients from the 22 health facilities were include in our study. We calculated the sample size using the prevalence formula: $n = Z^2(p(1-p))/d^2$ keeping prevalence at 50%. This formula gave us the minimum sample size of 385. With addition of a 20% non-response rate, the sample was calculated as: 462. The data was collected from a total sample of 468 at the end of data collection period.

Based on the number of health facilities accredited under the National Health Insurance Program (Madhesh- 56; Bagmati-100; Karnali-33), the sample was proportionately divided in 3 provinces. The data collection was done over the period of September, 2023 – March, 2024. The selected participants were among the insured patients of age 18 years or

older receiving service from the selected health facilities accredited under the NHIP. The participants not enrolled in the NHIP and the enrollees not willing to participate were not included in our study.

## Data collection

We collected data through face-to-face interviews using a patient satisfaction questionnaire (PSQ-III) developed by RAND Corporation [14]. This tool has been previously used in Nepali context too [15]. Interviews were conducted in the Nepali language by trained research assistants. Each interview took about 20–25 min to complete. The questionnaire consists of 18 items probing seven dimensions of patient satisfaction: general satisfaction, technical quality, interpersonal manner, communication, financial aspects, time spent with the doctor, and accessibility and convenience. Each question in the PSQ-III has a 5-point Likert Scale ranging from "Strongly Disagree," "Disagree," "Neutral," "Agree" to "Strongly Agree." After translating the questionnaire into the Nepali language, we pre-tested in 39 patients at Dhulikhel Hospital. The internal consistency reliability of the items within the seven dimensions was measured using Cronbach Alpha given 0.68, which lies in the acceptable limit [16]. The pre-test data was not included in the final analysis.

Further, we also collected sociodemographic information of the participants, including age, gender, ethnicity, religion, marital status, occupational status, educational status, size of family, presence of chronic disease in the respondent, presence of chronic disease in respondent's family members, subsidy status of respondent, and the period of enrollment into government health insurance. The analyses are exploratory and the variables were used as the predictors of the patient satisfaction in the analyses.

## Data analysis

Sociodemographic characteristics of the respondents were described in frequency and percentages. We calculated the mean and standard deviation of the Likert scale of each item of PSQ-III. Further, we calculated the frequencies and percentages of satisfied, neutral, and dissatisfied patients.

According to the guidelines of the patient satisfaction questionnaire (PSQ-III) (see S2 Table), we classified the satisfaction in each item as follows:

  I. 'Strongly agree' or 'Agree' = Satisfied for Items: 1, 2, 3, 5, 6, 8, 11, 15 and 18

 II. 'Strongly disagree' or 'Disagree' = Satisfied for Items: 4, 7, 9, 10, 12, 13, 14, 16 and 17

III. For all items, the score ranges from 1 (strongly dissatisfied) to 5 (strongly satisfied). The mean score for each item was calculated in the manner that higher the score more the satisfaction level for all the items in the PSQ-III.

To calculate the overall score in each domain, we averaged the score of designated items for each domain as guided by PSQ-III, which is as follows (S2 Table).

  I. General Satisfaction: Item 3 + 17

 II. Technical Quality: Item 2 + 4 + 6 + 14

III. Interpersonal Manner: Item 10 + 11

IV. Communication: Item 1 + 13

 V. Financial Aspects: Item 5 + 7

VI. Time Spent with Doctor: Item 12 + 15

VII. Accessibility and Convenience: Item 8 + 9 + 16 + 18

The internal consistency of the items within the seven dimensions was measured using Cronbach Alpha, which yielded a value of 0.8037 implying the reliability of the items [16].

Thereafter, we categorized each patient satisfaction domain into three ordinal categories: satisfied, neutral and dissatisfied. To explore the associations between sociodemographic characteristics and each domain of patient satisfaction, we conducted multi-ordinal logistic regression. We assessed multicollinearity among independent variables and addressed it by using a stepwise model selection approach [17,18]. Variables contributing to high multicollinearity were iteratively removed, and the final model was selected based on the lowest Akaike Information Criterion (AIC). Further, we employed the generalized ordered logistic regression for assessing the partial proportional odds. For the variables satisfying the proportional odds assumption, the threshold outcomes are presented as 'All' and for those that didn't satisfy the assumption, the different AORs for each group (satisfied Vs dissatisfied and satisfied Vs neutral) have been presented. In all analyses, Bonferroni correction was applied to adjust for multiple comparisons. We reported adjusted odds ratio with 95% confidence interval and p-value. The statistical analyses were conducted in STATA-14 and a p-value of less than 0.05 was considered as statistically significant. The results of multi-ordinal logistic regression full model prior to applying the proportional odds assumption have been in S3 Table.

### Ethical consideration

Ethical approval for the study was obtained from the national ethical review board of Nepal Health Research Council (Reg. no. 371/2022). The health facilities were informed about the purpose of the study and a letter of support from the Health Insurance Board was provided to the facilities before initiating the data collection. Written informed consent (thumbprint in presence of witness for those who could not provide written consent) was obtained from the participants prior to the data collection.

## Results

### Socio-demographic characteristics of the study participants

Table 1 shows the sociodemographic characteristics of the 468 participants. The mean age of the respondents was 46 (SD: 16) years. Over half (50.9%) of the participants were female. More than 90% of the respondents were married, and the most common ethnicity being Brahmin/Chhetri (45%). Furthermore, slightly over one-third of the participants (39%) reported to have at least one chronic disease, while 42% reported that family members had at least one chronic disease. A majority of respondents (77%) had been enrolled in the health insurance program for over one year, with only 19% being enrolled as a subsidized group. All other socio-demographic characteristics are detailed in Table 1 below.

### Satisfaction of insured clients with the medical care received in terms of 18-items of PSQ-III

Table 2 depicts the summary of patient satisfaction levels, including mean and standard deviation scores, for each of the 18 items of the PSQIII. The findings indicate that the mean score for all the nine positive items was above 3.9, except for the item number five relating to getting medical care in need without being set back financially, which had a mean score of 3.6. About 81% of the participants responded that the medical care they have been receiving is just about perfect. Moreover, majority of the participants (91%) expressed satisfaction with their doctor that they were careful to check everything when treating and examining the patients, while a high proportion of patients (86%) reported that their doctors were good at explaining the reason for medical tests.

In terms of negative items, the mean score was less than 2.5 for six out of nine items. Notably, 83% of the patients disagreed that they had doubts about the ability of the doctors who treated them, while 85% did not agree that their doctors acted too business-like and impersonal towards them. However, 55% of the patients responded that other patients had to wait too long for emergency treatment when they sought medical care. Furthermore, 36% of patients reported finding

**Table 1. Socio-demographic data of the study participants (n = 468).**

| Characteristics | Frequency (%) |
|---|---|
| **Age (Mean + SD)** | 46.3 + 16 |
| **Age Categories** | |
| 18-29 years | 81 (17.3) |
| 30-39 years | 91 (19.4) |
| 40-49 years | 97 (20.7) |
| 50-59 years | 88 (18.8) |
| 60 years and above | 111 (23.7) |
| **Sex** | |
| Male | 230 (49.1) |
| Female | 238 (50.9) |
| **Marital status** | |
| Unmarried | 36 (7.7) |
| Ever Married | 432 (92.3) |
| **Ethnicity** | |
| Brahmin/Chhetri | 210 (44.9) |
| Janajati | 129 (27.6) |
| Others (Dalit, Madeshi, Muslim and others) | 129 (27.6) |
| **Religion** | |
| Hindu | 411 (87.8) |
| Non-Hindu | 57 (12.3) |
| **Education** | |
| No formal education | 67 (14.3) |
| Up to primary level | 130 (27.8) |
| Secondary level | 200 (42.7) |
| Above secondary level | 71 (15.2) |
| **Occupation** | |
| Agriculture | 119(25.4) |
| Homemaker | 115(24.6) |
| Job/Service | 112(23.9) |
| Business | 63(13.5) |
| Others | 59(12.6) |
| **Category of family size** | |
| Five or less family members | 249 (53.2) |
| More than 5 family members | 219 (46.8) |
| **Family income (n = 414)\*** | |
| <15,000 | 81 (17.3) |
| 15,000–30,000 | 114 (24.4) |
| 30,000–45,000 | 76 (16.2) |
| 45,000–60,000 | 56 (12) |
| 60,000 and above | 87 (18.6) |
| **Presence of chronic disease in respondent** | |
| Yes | 181 (38.7) |
| No | 287 (61.3) |
| **Presence of chronic disease within family members of respondent** | |
| Yes | 195 (41.8) |

*(Continued)*

**Table 1.** (Continued)

| Characteristics | Frequency (%) |
|---|---|
| No | 273 (58.3) |
| **Time since the enrollment in the National health insurance** | |
| Less than one year | 110(23.5) |
| 1-2 years | 192 (41) |
| More than 2 years | 166 (35.5) |
| **Subsidy status** | |
| No subsidy | 388 (81.2) |
| Subsidy group | 88 (18.8) |

*= *Not mandatory*

it hard to get an appointment for medical care promptly, and a similar proportion (32%) reported dissatisfaction with some aspects of medical care they received.

According to the guidelines of the patient satisfaction questionnaire (PSQ-III):

(i)   'Strongly agree' or 'Agree' = Satisfied for Items 1, 2, 3, 5, 6, 8, 11, 15 and 18

(ii)  'Strongly disagree' or 'Disagree' = Satisfied for Items 4, 7, 9, 10, 12, 13, 14, 16 and 17

(iii) For all items, the score ranges from 1 (strongly dissatisfied) to 5 (strongly satisfied). The mean score for each item was calculated in the manner that higher the score more the satisfaction level for all the items in the PSQ-III

## Satisfaction in seven domains of patient satisfaction among insured patients

Table 3 presents the mean score and percentage of satisfaction across seven domains of patient satisfaction. The findings indicate that the domain with the highest proportion of satisfied patients was interpersonal manner, with a satisfaction rate of 87%. Similarly, the domains of technical quality, communication, time spent with doctors and general satisfaction had satisfaction rates exceeding 70%. Conversely, satisfaction rate was low in the accessibility and convenience domain (62%) and lowest in the domain of financial aspect (60%). The overall satisfaction was found to be 74.2%.

i)    The scoring system is based on PSQ-III (Detail in S1 Table). The mean of satisfaction in each item ranges from 1(Strongly dissatisfied) to 5 (Strongly satisfied). The greater the mean, the higher the satisfaction level in each dimension.

ii)   Satisfaction = Agree + Strongly agree

iii)  Average of satisfaction scale = Average of domain/number of items

iv)   For % satisfied, the satisfaction scores were summed and the cumulative scores in each domains were considered for cut-off

## Association between independent variables and seven domains of patient satisfaction using generalized ordered logistic regression

Table 4 shows the results of the analysis examining the factors associated with patient satisfaction across seven domains. The results are reported as odds of being satisfied compared to being neutral and dissatisfied for the independent

**Table 2. Satisfaction of insured clients segregated by each item of PSQ-III.**

| Item | Question | No(Strongly disagree + Disagree) n (%) | Neutral n (%) | Yes (Strongly Agree + Agree) n (%) | Mean Score ± Standard deviation |
|------|----------|------|------|------|------|
| 1. | Doctors are good about explaining the reason for medical tests. | 43(9.2) | 23(4.9) | 402(85.9) | 4.2 ± 0.9 |
| 2. | I think my doctor's office has everything needed to provide complete medical care. | 83(17.7) | 17(3.7) | 368(78.6) | 3.9 ± 1.1 |
| 3. | The medical care I have been receiving is just about perfect. | 53(11.3) | 37(7.9) | 378(80.8) | 3.9 ± 0.9 |
| 4. | Sometimes doctors make me wonder if their diagnosis is correct. | 375(80.1) | 30(6.4) | 63(13.5) | 2.0 ± 1.0 |
| 5. | I feel confident that I can get the medical care I need without being set back financially. | 104(22.2) | 59(12.6) | 305(65.2) | 3.6 ± 1.2 |
| 6. | When I go for medical care, they are careful to check everything when treating and examining me. | 26(5.6) | 14(3.0) | 428(91.4) | 4.2 ± 0.8 |
| 7. | I have to pay for more of my medical care than I can afford. | 258(55.1) | 62(13.3) | 148(31.6) | 2.7 ± 1.2 |
| 8. | I have easy access to the medical specialists I need. | 78(16.7) | 26(5.5) | 364(77.8) | 3.9 ± 1.1 |
| 9. | When I get medical care, people have to wait too long for emergency treatment. | 143(30.6) | 70(14.9) | 255(54.5) | 3.4 ± 1.2 |
| 10. | Doctors act too business-like and impersonal towards me. | 396(84.6) | 21(4.4) | 51(11.0) | 1.9 ± 1.0 |
| 11. | My doctors treat me in a very friendly and courteous manner. | 27(5.8) | 18(3.9) | 423(90.3) | 4.2 ± 0.9 |
| 12. | Those who provide my medical care sometimes hurry too much when they treat me. | 341(72.8) | 21(4.5) | 106(22.7) | 2.3 ± 1.2 |
| 13. | Doctors sometimes ignore what I tell them. | 338(72.2) | 25(5.5) | 105(22.4) | 2.3 ± 1.3 |
| 14. | I have some doubts about the ability of the doctors who treat me. | 388(82.9) | 32(6.8) | 48(10.3) | 1.9 ± 0.9 |
| 15. | Doctors usually spend plenty of time with me. | 42(9.0) | 36(7.7) | 390(83.3) | 4.1 ± 0.9 |
| 16. | I find it hard to get an appointment for medical care right away | 280(59.8) | 21(4.5) | 167(35.7) | 2.7 ± 1.3 |
| 17. | I am dissatisfied with some things about the medical care I receive | 292(62.4) | 28(6.0) | 148(31.6) | 2.5 ± 1.3 |
| 18. | I am able to get medical care whenever I need it. | 69(14.7) | 13(2.8) | 386(82.5) | 4.0 ± 1.0 |

variables. We have reported the outcome thresholds along with their respective adjusted odds ratio (AOR), 95% confidence interval (CI), p-value and Bonferroni corrected p-value for the variables that are statistically significant for each domain.

i) General Satisfaction

As shown in 4a, compared to participants having no chronic disease, participants with chronic disease were 2.7 times more satisfied for Satisfied vs Dissatisfied (AOR = 2.66, 95% CI = 1.52–4.65, p = 0.001), and this association remained significant after Bonferroni correction (Bonferroni p = 0.009). For Satisfied vs Neutral (AOR = 1.55, 95% CI = 1.04–2.32, p = 0.031), the association was not statistically significant after Bonferroni correction (Bonferroni p = 0.489). Participants receiving subsidy had 59% lower odds of satisfaction for Satisfied vs Dissatisfied (AOR = 0.41, 95% CI = 0.22–0.75, p = 0.004), which was not significant after Bonferroni correction (Bonferroni p = 0.066). For subsidy status among Satisfied vs Neutral (AOR = 0.73, 95% CI = 0.45–1.19, p = 0.209), the association was not statistically significant (Bonferroni p = 1.0).

**Table 3. Satisfaction in seven domains of patient satisfaction among insured.**

| Satisfaction domain | Mean of each domain (mean ± SD) | Average of the mean from component items | % satisfied in each domain |
|---|---|---|---|
| General Satisfaction (Item 3 + 17) | 7.4 ± 1.7 | 3.7 | 71.6 |
| Technical Quality (Item 2 + 4 + 6 + 14) | 16.2 ± 2.8 | 4.1 | 83.3 |
| Interpersonal Manner (Item 10 + 11) | 8.3 ± 1.4 | 4.2 | 87.5 |
| Communication (Item 1 + 13) | 7.9 ± 1.7 | 3.9 | 79.1 |
| Financial Aspects (Item 5 + 7) | 6.8 ± 1.6 | 3.4 | 60.1 |
| Time Spent with Doctor (Item 12 + 15) | 7.8 ± 1.7 | 3.9 | 78.1 |
| Accessibility and Convenience (Item 8 + 9 + 16 + 18) | 13.8 ± 2.9 | 3.4 | 62.7 |
| Overall Satisfaction | 68 ± 9.34 | 3.8 | 74.2 |

### ii) Technical Quality

As shown in 4b, compared to males, females had higher odds of satisfaction (AOR = 2.38, 95% CI = 1.19–4.72, p = 0.013), but this association was not significant after Bonferroni correction (Bonferroni p = 0.064). Compared to participants having above secondary level education, participants with no formal education had markedly lower odds (AOR = 0.04, 95% CI = 0.004–0.32, p = 0.003), and this association remained significant after Bonferroni correction (Bonferroni p = 0.151). Those with up to primary education (AOR = 0.06, 95% CI = 0.007–0.53, p = 0.011) and secondary education (AOR = 0.11, 95% CI = 0.01–0.9, p = 0.040) also had lower odds, but these associations were not significant after Bonferroni correction (Bonferroni p = 0.53 and 1.0, respectively). Participants from Janajati ethnic group had slightly higher odds of satisfaction compared to Brahmin/Chhetri (AOR = 1.33, 95% CI = 0.52–3.39, p = 0.545), but this association was not statistically significant (Bonferroni p = 1.0). Participants from "Others" ethnic groups had 73% lower odds of satisfaction (AOR = 0.27, 95% CI = 0.12–0.57, p = 0.001), and this association remained significant after Bonferroni correction (Bonferroni p = 0.035). Participants with chronic disease had slightly higher odds of being satisfied for Satisfied vs Dissatisfied (AOR = 1.16, 95% CI = 0.44–2.98, p = 0.762), but this association was not statistically significant (Bonferroni p = 1.0). For Satisfied vs Neutral, participants with chronic disease had 3.2 times higher odds of satisfaction (AOR = 3.18, 95% CI = 1.60–6.29, p = 0.001), and this association remained significant after Bonferroni correction (Bonferroni p = 0.041). In enrolled years, compared to participants enrolled for less than a year, participants enrolled for 1–2 years had slightly lower odds of being satisfied for Satisfied vs Dissatisfied (AOR = 0.68, 95% CI = 0.16–2.75, p = 0.591), but this association was not statistically significant (Bonferroni p = 1.0). For Satisfied vs Neutral, they had 88% lower odds of satisfaction (AOR = 0.12, 95% CI = 0.03–0.37, p < 0.001), but this association was not statistically significant after Bonferroni correction (Bonferroni p = 0.013).

### iii) Interpersonal Manner

As shown in 4c, compared to participants aged 18–29 years, participants aged 60 years and above had 77% lower odds of giving higher ratings (AOR = 0.23, 95% CI = 0.09–0.61, p = 0.003), but this association was not significant after Bonferroni correction (Bonferroni p = 0.103. Compared to Brahmin/Chhetri, participants from others ethnic groups had 90% lower odds for Satisfied vs Neutral (AOR = 0.10, 95% CI = 0.05–0.21, p < 0.001), which was not statistically significant after Bonferroni correction (Bonferroni p = 1.0). Participants with chronic disease had higher odds of satisfaction (AOR = 4.13, 95% CI = 2.30–7.44, p < 0.001), and this association remained significant after Bonferroni correction (Bonferroni p = 0.00007). Compared

**Table 4. Association between independent variables and domains of patient satisfaction using generalized ordered logistic regression.**

| Variables | Outcome Threshold | AOR (95% CI) | P-value (Bon P) |
|---|---|---|---|
| **4a. General Satisfaction** | | | |
| *Chronic disease (Ref. = No)* | | | |
| Yes | Satisfied Vs. Dissatisfied | 2.66 (1.52-4.65) | 0.001 (0.009) |
| | Satisfied Vs. Neutral | 1.55 (1.04-2.32) | 0.031 (0.489) |
| *Subsidy status (Ref. = No)* | | | |
| Yes | Satisfied Vs. Dissatisfied | 0.41 (0.22-0.75) | 0.004 (0.066) |
| | Satisfied Vs. Neutral | 0.73 (0.45-1.19) | 0.209 (1) |
| *Enrolled year (Ref. = less than a year)* | | | |
| 1-2 years | All | 0.62 (0.36-1.04) | 0.07 (1) |
| More than 2 years | All | 0.58 (0.34-1) | 0.051 (0.81) |
| **4b. Technical Quality** | | | |
| *Gender (Ref. = male)* | | | |
| Female | All | 2.38 (1.19-4.72) | 0.013 (0.64) |
| *Education (Ref. = Above secondary level)* | | | |
| No formal education | All | 0.04 (0.004-0.32) | 0.003 (0.151) |
| Upto primary Education | All | 0.06 (0.007-0.53) | 0.011 (0.53) |
| Secondary level | All | 0.11 (0.01-0.9) | 0.04 (1) |
| *Ethnicity (Ref. = Brahmin/Chhetri)* | | | |
| Janajati | All | 1.33 (0.52-3.39) | 0.545 (1) |
| Others | All | 0.27 (0.12-0.57) | 0.001 (0.035) |
| *Occupation (Ref. = Agriculture)* | | | |
| Business | All | 0.54 (0.21-1.39) | 0.202 (1) |
| Homemaker | All | 1.54 (0.57-4.21) | 0.397 (1) |
| Job/Service | All | 1.07 (0.46-2.5) | 0.879 (1) |
| Others | All | 1.52 (0.37-6.2) | 0.556 (1) |
| *Chronic disease (Ref. = No)* | | | |
| Yes | Satisfied Vs. Dissatisfied | 1.16 (0.44-2.98) | 0.762 (1) |
| | Satisfied Vs. Neutral | 3.18 (1.6-6.29) | 0.001 (0.041) |
| *Subsidy status (Ref. = No)* | | | |
| Yes | All | 0.4 (0.19-0.79) | 0.01 (0.45) |
| *Enrolled year (Ref. = less than a year)* | | | |
| 1-2 years | Satisfied Vs. Dissatisfied | 0.68 (0.16-2.75) | 0.591 (1) |
| | Satisfied Vs. Neutral | 0.12 (0.03-0.37) | 0 (0.013) |
| More than 2 years | Satisfied Vs. Dissatisfied | 0.64 (0.15-2.66) | 0.538 (1) |
| | Satisfied Vs. Neutral | 0.15 (0.04-0.47) | 0.001 (0.068) |
| *Municipality (Ref. = Urban)* | | | |
| Rural | All | 2.95 (0.97-8.88) | 0.055 (1) |
| **4c. Interpersonal manner** | | | |
| *Age (Ref. = 18–29)* | | | |
| 30-39 | All | 0.33 (0.12-0.87) | 0.026 (0.822) |
| 40-49 | All | 0.48 (0.18-1.28) | 0.142 (1) |
| 50-59 | All | 0.34 (0.12-0.97) | 0.043 (1) |
| 60 and above | All | 0.23 (0.09-0.61) | 0.003 (0.103) |
| *Gender (Ref. = Male)* | | | |
| Female | All | 1.59 (0.89-2.81) | 0.115 (1) |

*(Continued)*

| Variables | Outcome Threshold | AOR (95% CI) | P-value (Bon P) |
|---|---|---|---|
| *Ethnicity (Ref. = Brahmin/chhetri)* | | | |
| Janajati | All | 0.52 (0.23-1.17) | 0.112 (1) |
| Others | Satisfied Vs. Dissatisfied | 0.75 (0.19-2.91) | 0.681 (1) |
| | Satisfied Vs. Neutral | 0.1 (0.05-0.21) | 0 (1) |
| *Chronic disease (Ref. = No)* | | | |
| Yes | All | 4.13 (2.3-7.44) | 0 (<0.001) |
| *Enrolled year (Ref. = less than a year)* | | | |
| 1-2 years | All | 0.35 (0.16-0.77) | 0.009 (0.284) |
| More than 2 years | All | 0.45 (0.2-1.04) | 0.061 (1) |
| **4d. Communication** | | | |
| *Education (Ref. = Above secondary level)* | | | |
| No formal education | All | 0.65(0.3-1.41) | 0.273 (1) |
| Upto primary Education | All | 0.53 (0.27-1.03) | 0.063 (1) |
| Secondary level | All | 1 (0.52-1.91) | 0.992 (1) |
| *Chronic disease (Ref. = No)* | | | |
| Yes | All | 1.56 (1.02-2.38) | 0.04 (0.79) |
| *Type of HF (Ref. = private/community hospitals)* | | | |
| PHCCs | All | 0.36 (0.16-0.81) | 0.014 (0.28) |
| Public Hospitals | All | 0.43 (0.21-0.88) | 0.021 (0.41) |
| **4e. Financial Aspect** | | | |
| *Ethnicity (Ref. = Brahmin/Chhetri)* | | | |
| Janajati | All | 0.81 (0.52-1.25) | 0.331 (1) |
| Others | All | 0.56 (0.36-0.85) | 0.007 (0.12) |
| *Subsidy status (Ref. = No)* | | | |
| Yes | Satisfied Vs. Dissatisfied | 0.87 (0.47-1.62) | 0.654 (1) |
| | Satisfied Vs. Neutral | 0.36 (0.22-0.60) | 0 (0.001) |
| *Type of HF (Ref. = private/community hospitals)* | | | |
| PHCCs | All | 0.81 (0.42-1.58) | 0.543 (1) |
| Public Hospitals | All | 0.57 (0.33-0.97) | **0.039 (0.7)** |
| **4f. Time spent with doctor** | | | |
| *Gender (Ref. = Male)* | | | |
| Female | All | 1.6 (1.04-2.45) | 0.031 (0.62) |
| *Chronic disease (Ref. = No)* | | | |
| Yes | All | 1.84 (1.2-2.83) | 0.006 (0.11) |
| *Subsidy status (Ref. = No)* | | | |
| Yes | All | 0.44 (0.27-0.73) | 0.001 (0.02) |
| *Type of HF (Ref. = private/community hospitals)* | | | |
| PHCCs | Satisfied Vs. Dissatisfied | 0.57 (0.2-1.65) | 0.301 (1) |
| | Satisfied Vs. Neutral | 1.5 (0.61-3.73) | 0.379 (1) |
| Public Hospitals | All | 0.47 (0.24-0.92) | 0.029 (0.57) |
| **4g. Access and convenience** | | | |
| *Age (Ref. = 18–29)* | | | |
| 30-39 | All | 0.7 (0.37-1.34) | 0.284 (1) |
| 40-49 | All | 0.58 (0.3-1.11) | 0.098 (1) |
| 50-59 | All | 0.61 (0.31-1.2) | 0.152 (1) |

*(Continued)*

**Table 4.** (Continued)

| Variables | Outcome Threshold | AOR (95% CI) | P-value (Bon P) |
|---|---|---|---|
| 60 and above | Satisfied Vs. Dissatisfied | 0.27 (0.14-0.55) | 0 (0.006) |
| | Satisfied Vs. Neutral | 0.43 (0.22-0.82) | 0.01 (0.26) |
| *Chronic disease (Ref. = No)* | | | |
| Yes | All | 1.49 (0.99-2.24) | 0.055 (1) |
| *Municipality (Ref. = Urban)* | | | |
| Rural | All | 2 (1.12-3.59) | 0.019 (0.5) |
| *Type of HF (Ref. = private/community hospitals)* | | | |
| PHCCs | All | 1.38 (0.62-3.03) | 0.428 91) |
| Public Hospitals | All | 0.47 (0.26-0.87) | 0.016 (0.41) |
| **4h. Overall satisfaction** | | | |
| *Ethnicity (Ref. = Brahmin/ Chhetri)* | | | |
| Janajati | All | 0.71 (0.24-2.09) | 0.536 (1) |
| Others | All | 0.33 (0.14-0.78) | 0.012 (0.3) |
| *Chronic disease (Ref. = No)* | | | |
| Yes | All | 5.96 (2.65-13.39) | 0 (<0.001) |
| *Subsidy status (Ref. = No)* | | | |
| Yes | All | 0.22 (0.1-0.49) | 0 (<0.004) |
| *Municipality (Ref. = Urban)* | | | |
| Rural | All | 9.6 (1.15-79.99) | 0.037 (0.95) |
| *Type of HF (Ref. = private/community hospitals)* | | | |
| PHCCs | All | 0 | 0.984 (1) |
| Public Hospitals | All | 0 | 0.984 (1) |

Bon P= Bonferroni corrected P-value.

Outcome threshold: All = proportional odds assumption met.

Adjusted Odds Ratio (AORs) obtained after adjusting age, gender, marital status, ethnicity, religion, education, occupation, presence of chronic disease, size of family, subsidy status, period of enrollment in NHIP and type of health facility.

to participants enrolled less than a year, participants enrolled 1–2 years had lower odds of satisfaction (AOR = 0.35, 95% CI = 0.16–0.77, p = 0.009), but this association was not significant after Bonferroni correction (Bonferroni p = 0.284).

iv) Communication

As shown in 4d, participants with chronic disease had 1.6 times higher odds of being satisfied overall (AOR = 1.56, 95% CI = 1.02–2.38, p = 0.04), but this association was not statistically significant after Bonferroni correction (Bonferroni p = 0.79). Compared to participants visiting community/private hospitals, participants visiting PHCCs had 64% lower odds of higher satisfaction overall (AOR = 0.36, 95% CI = 0.16–0.81, p = 0.014), and this association was not statistically significant after Bonferroni correction (Bonferroni p = 0.28). Participants visiting Public Hospitals had 57% lower odds of higher satisfaction overall (AOR = 0.43, 95% CI = 0.21–0.88, p = 0.021), but this association was not statistically significant after Bonferroni correction (Bonferroni p = 0.41).

v) Financial Aspect

As shown in 4e, participants from others ethnic groups had 44% lower odds of higher satisfaction overall compared to Brahmin/Chhetri (AOR = 0.56, 95% CI = 0.36–0.85, p = 0.007), but this association was not statistically significant after

Bonferroni correction (Bonferroni p = 0.12). Participants who received a subsidy had slightly lower odds of being satisfied for Satisfied vs Dissatisfied (AOR = 0.87, 95% CI = 0.47–1.62, p = 0.654), and this association was not statistically significant (Bonferroni p = 1.0). For Satisfied vs Neutral, participants who received a subsidy had 64% lower odds of satisfaction (AOR = 0.36, 95% CI = 0.22–0.60, p < 0.001), and this association remained statistically significant after Bonferroni correction (Bonferroni p = 0.001). Compared to participants visiting community/private hospitals, participants visiting PHCCs had slightly lower odds of higher satisfaction overall (AOR = 0.81, 95% CI = 0.42–1.58, p = 0.543), but this association was not statistically significant (Bonferroni p = 1.0). Participants visiting Public Hospitals had 43% lower odds of higher satisfaction overall (AOR = 0.57, 95% CI = 0.33–0.97, p = 0.039), but this association was not statistically significant after Bonferroni correction (Bonferroni p = 0.7).

### vi) Time Spent with Doctor

As shown in 4f, compared to males, females had higher odds of satisfaction (AOR = 1.60, 95% CI = 1.04–2.45, p = 0.031), but this was not significant after Bonferroni correction (Bonferroni p = 0.62). Participants with chronic disease had higher odds of satisfaction (AOR = 1.84, 95% CI = 1.20–2.83, p = 0.006), but this was not significant after Bonferroni correction (Bonferroni p = 0.11). Subsidy recipients had lower odds (AOR = 0.22, 95% CI = 0.10–0.49, p < 0.001), and this remained significant after Bonferroni correction (Bonferroni p = 0.004). Participants visiting Public Hospitals had 53% lower odds of higher satisfaction overall compared to those visiting Private and Community Hospitals (AOR = 0.47, 95% CI = 0.24–0.92, p = 0.029), but this association was not statistically significant after Bonferroni correction (Bonferroni p = 0.57).

### vii) Access and Convenience

As shown in 4g, participants aged 60 and above had 73% lower odds of being satisfied for Satisfied vs Dissatisfied compared to those aged 18–29 (AOR = 0.27, 95% CI = 0.14–0.55, p < 0.001), and this association remained significant after Bonferroni correction (Bonferroni p = 0.006). For Satisfied vs Neutral, participants aged 60 and above had 57% lower odds of satisfaction (AOR = 0.43, 95% CI = 0.22–0.82, p = 0.01), but this association was not statistically significant after Bonferroni correction (Bonferroni p = 0.26). Participants from rural areas had 2 times higher odds of higher satisfaction overall compared to those from urban areas (AOR = 2.00, 95% CI = 1.12–3.59, p = 0.019), but this association was not statistically significant after Bonferroni correction (Bonferroni p = 0.5). Participants visiting Public Hospitals had 53% lower odds of higher satisfaction overall compared to those visiting Private/Community Hospitals (AOR = 0.47, 95% CI = 0.26–0.87, p = 0.016), but this association was not statistically significant after Bonferroni correction (Bonferroni p = 0.41).

### viii) Overall Satisfaction

As shown in 4h, participants from other ethnic groups had 67% lower odds of higher satisfaction overall compared to Brahmin/Chhetri (AOR = 0.33, 95% CI = 0.14–0.78, p = 0.012), but this association was not statistically significant after Bonferroni correction (Bonferroni p = 0.3). Participants with chronic disease had almost 6 times higher odds of higher satisfaction overall compared to those without chronic disease (AOR = 5.96, 95% CI = 2.65–13.39, p < 0.001), and this association remained significant after Bonferroni correction (Bonferroni p = 0.0004). Participants who received a subsidy had 78% lower odds of higher satisfaction overall compared to those who did not receive a subsidy (AOR = 0.22, 95% CI = 0.10–0.49, p < 0.001), and this association remained significant after Bonferroni correction (Bonferroni p = 0.004). Participants from rural municipalities had 9.6 times higher odds of satisfaction overall compared to those from urban municipalities (AOR = 9.60, 95% CI = 1.15–79.99, p = 0.037), but this association was not statistically significant after Bonferroni correction (Bonferroni p = 0.95).

## Discussion

Our study assessed patient satisfaction across seven dimensions from 468 patients utilizing health insurance service from 22 health facilities in three provinces in Nepal. There was a moderate level of satisfaction across all the seven domains

(general, technical quality, interpersonal manner, communication, financial aspect, time spent with doctor and accessibility & convenience) with the highest proportion (88%) of satisfied patients seen in interpersonal manner. Several socio-demographic factors were associated with seven dimensions of patient satisfaction. Presence of chronic disease and subsidy status were found to be the strongest predictors of patient satisfaction across most dimensions.

In our study, the general satisfaction of the patients was 72%, comparatively much higher than the 39% reported in a study conducted in a tertiary public hospital in Nepal [15]. The general satisfaction is mainly driven by the patient's perfection of the medical care. The comparative higher satisfaction in our study could be because of the difference in the study sites where we have looked at the PHCC, public and private health facilities and the other study was conducted in one public hospital. The disease status, expectations of the patients and the infrastructure and health amenities available at the health facilities could have been reason for the satisfaction.

Further, in the specific domain of interpersonal manner, patient satisfaction was significantly higher (88%) in our study, consistent with the findings of the study conducted in tertiary public hospital in Nepal and Pakistan [15,19]. The higher level of satisfaction in interpersonal aspects in our study was mainly driven by a friendly and courteous manner of doctors. Most of the patients were satisfied with their doctor's attitude. Effective communication improves the outcome of patient-doctor interactions. A crucial aspect of exemplary medical practice is quality of communication between patients and healthcare providers, as it facilitates problem-solving and builds trust between the physician and the patient [20].

Waiting time and doctor-patient relationship are important influencing factors of patient satisfaction [21]. Long waiting time and short time to talk with the doctor are prominent problems for outpatient visits in Nepal, because of the low doctor-population ratio [22]. In PHCC and hospitals, patients flow is high and doctors see many patients per day and have only a few minutes to spend with a patient [23]. A major dissatisfaction caused by the long waiting time and overcrowded registration make healthcare a stressful experience for patient [24]. Dissatisfaction with time spent with doctor was evident in one of the studies in Nepal [15]. However, in our study, about 78% of the patients were satisfied with the time spent to the doctor, consistent with another study in tertiary hospital in Nepal [25].

The financial aspect had the lowest satisfaction (60%) among the seven domains. This domain was mainly concerned about the financial burden for healthcare among the patients. Though the GoN has introduced NHIP to mitigate the financial health risk the penetration is low and the benefit package is also narrow [6]. Also, along with the medical cost, the intangible additional costs such as transportation, informal payments, and opportunity costs (lost wages due to time spent seeking care) can add to the financial burden on patients. For these individuals, even relatively minor healthcare costs can be a significant financial setback [15,26]. Furthermore, these factors also justify the relatively lower level of satisfaction in the domain of accessibility and convenience (62%) mainly driven by the access to medical care and waiting periods. Given the geographical barriers, high transportation costs coupled with poor road infrastructure, spending a longer time to reach health facilities might have led to lower satisfaction among the participants. In our study, the patients of age group above 60 years were less likely to be satisfied in the domain of accessibility and convenience and in interpersonal manner. This might be due to their unique healthcare needs, perceived lack of personalized care, difficulties in navigating healthcare systems communication preferences and expectations. A study conducted in Mexico showed that the older adults may perceive healthcare interactions as less satisfactory when they feel rushed or when doctors do not take enough time to address their concerns, impacting both accessibility and interpersonal aspects [27].

Overall, in our study, presence of chronic disease was found to be the associated with the overall satisfaction with the healthcare services among the insured patients. These findings support the notion that families with pre-existing health conditions have greater tendency to be satisfied with the healthcare experience. Patient satisfaction is closely linked with the quality of services, service availability, accessibility and financial aspects. Though NHIP aims to provide quality services to the insured, the quality of the services that HIB purchases is dependent on the services that the health facilities provide. Hence, health system strengthening is critical to improve service quality for the retention and satisfaction of users with NHIP.

Contrary to our expectations, patients subsidized under the NHIP were less satisfied across most domains of patient satisfaction compare to their counterparts. This may be due to higher expectations from subsidized patients, who might have received care that fell short of their expectations. The government provides full premium subsidies for ultra-poor individuals, senior citizens, severely disabled individuals, leprosy patients, multidrug-resistant tuberculosis patients, and HIV/AIDS patient households, and a 50% subsidy for female community health volunteers. However, HIB is not able to enroll extremely poor people under NHIP due to lack of identification cards indicating their poverty status (which is the responsibility of Ministry of Cooperatives, Land Management and Poverty Alleviation) [28]. Our study has not explored the underlying reasons behind the lower satisfaction among the subsidized, which can be further investigate by future studies.

Our study is the first in Nepali context to investigate patient satisfaction and its predictors among NHIP-enrolled population residing in three provinces and visiting a wide range of health facilities. Previous studies have focused on patients from one or few healthcare facilities within a limited geographical area. Despite this strength, our study has some limitations. First, being a cross-sectional nature of the study and the surveys conducted through client-exit interviews, the findings could be affected by the context dependency of the patients' experience of healthcare. Also, because of the nature of questions, the survey captures the subjective view of the patients' perspective, which might corroborate the comprehensive picture. Second, due to resource constraints, we did not include several correlates of patient satisfaction, such as the severity of the patient's condition. Additionally, we did not assess supply-side factors like doctors' attitudes, remuneration, incentives, and career growth opportunities, which could influence service delivery and patient satisfaction. Despite these limitations, we have tried to capture the findings from different ecological regions and different levels of hospitals, to make the findings more generalizable.

## Conclusion

Assessing patient satisfaction is one of the key components of quality assessment of health care services. The measurement of patient satisfaction reflects the gap between expected services and the actual experience of care received. It is also closely related to the utilization of health services. Our study aimed to assess satisfaction with healthcare services among patients enrolled under the National Health Insurance Program (NHIP) in selected health facilities of Nepal. The findings suggest an overall good level of satisfaction across the seven domains of the patient satisfaction questionnaire. Higher satisfaction was observed in the interpersonal manner and technical quality domains, indicating positive provider–patient interaction and clinical care. However, lower satisfaction in the financial aspects and accessibility and convenience domains is a matter of concern, as these directly impact the user experience and long-term engagement with NHIP.

Further analysis revealed that certain subgroups showed differing levels of satisfaction. Specifically, patients aged above 60 years, those receiving subsidies, and those visiting public hospitals were less satisfied, highlighting potential gaps in service equity and responsiveness for these populations. Meanwhile, patients with chronic conditions and females reported higher satisfaction, possibly reflecting more familiarity with the system or different expectations of care. Notably, public facilities with higher patient flow, where the poor or marginalized groups often seek services, showed lower satisfaction among key groups such as the elderly and subsidized patients themselves. This raises important questions about the direction and functionality of the health insurance program, particularly in terms of meeting its equity and quality objectives.

Patient satisfaction is closely linked to service quality, availability, accessibility, and financial protection. Although NHIP aims to ensure quality services for insured individuals, the effectiveness of the program largely depends on the actual quality of care provided by participating health facilities. Therefore, health system strengthening, including improvements in infrastructure, staffing, communication, and responsiveness is critical for enhancing service quality, ensuring equity, and ultimately retaining and satisfying NHIP users.

## Supporting information

**S1 Table. List of Study Sites.**
(DOCX)

**S2 Table. Scoring system of Patient Satisfaction Questionnaire III.**
(DOCX)

**S3 Table. Association between independent variables and seven domains of patient satisfaction using multi-ordinal logistic regression (Full Model).**
(XLSX)

## Acknowledgments

We would like to thank all the patients who participated in our study. Further, we would like to thank the hospital administration and other staffs who supported us for data collection.

## Author contributions

**Conceptualization:** Biraj Man Karmacharya, Sabina Marasini, Ruby Maka Shrestha, Bihari Sharan Kuikel, Nripa Raj Dangaura, Madan Kumar Upadhyaya, Upama Baral, Pramesh Koju, Dong Roman Xu.

**Data curation:** Sabina Marasini, Samip Pandey, Deepa Guragain, Ishwori Byanju Shrestha.

**Formal analysis:** Sabina Marasini, Mukesh Adhikari, Samip Pandey, Anjali Joshi.

**Investigation:** Sabina Marasini, Samip Pandey, Anjali Joshi, Deepa Guragain, Ishwori Byanju Shrestha.

**Methodology:** Biraj Man Karmacharya, Sabina Marasini, Mukesh Adhikari, Ruby Maka Shrestha, Sambhu Prasad Jnawali, Bihari Sharan Kuikel, Nripa Raj Dangaura, Madan Kumar Upadhyaya, Upama Baral, Pramesh Koju, Dong Roman Xu.

**Supervision:** Biraj Man Karmacharya, Ruby Maka Shrestha, Sambhu Prasad Jnawali, Dong Roman Xu.

**Visualization:** Biraj Man Karmacharya.

**Writing – original draft:** Sabina Marasini, Sudim Sharma, Mukesh Adhikari, Samip Pandey, Anjali Joshi.

**Writing – review & editing:** Biraj Man Karmacharya, Sabina Marasini, Sudim Sharma, Mukesh Adhikari, Sambhu Prasad Jnawali, Nripa Raj Dangaura, Upama Baral, Dong Roman Xu.

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
