## [Decision Letter · Decision Letter 0]

11 Jul 2025

Dear Dr. Marasini,

We look forward to receiving your revised manuscript.

Kind regards,

Kshitij Karki, MPH, MA

Academic Editor

PLOS ONE

Journal Requirements:

“The study was conducted with the funding support from Swiss Agency for Development and Cooperation (SDC). However, the SDC provided no direct funding for preparation and submission of this article.”

4. We noted in your submission details that a portion of your manuscript may have been presented or published elsewhere. “The dataset (socio-demographic characteristics) used in this manuscript has also been used in a separate publication (DOI:10.1186/s12913-025-12893-5). However, the analysis presented here is entirely distinct. This study focuses on the satisfaction with healthcare services whereas the other examined satisfaction with health insurance components.

The socio-demographic characteristics of respondents are the same due to the use of the same dataset, but the research questions, analytical approach, and findings are different. This manuscript has not been published or submitted elsewhere for publication.”

5. Please remove all personal information, ensure that the data shared are in accordance with participant consent, and re-upload a fully anonymized data set.

Additional guidance on preparing raw data for publication can be found in our Data Policy (https://journals.plos.org/plosone/s/data-availability#loc-human-research-participant-data-and-other-sensitive-data) and in the following article: http://www.bmj.com/content/340/bmj.c181.long .

**Additional Editor Comments:**

Thank you for the paper on patient satisfaction related to NHIP. Please revise the manuscript as per the reviewers suggestions.

In addition, please revise the sentence from 402 and 444 as they are similar. The recommendations in the conclusion should be specific based on the results. Also, clarify how did you define category of family members. In sample, how did you select the sample proportionately from the selected health facilities enrolled in HIB. Have you selected the health facilities randomly or what? Proportionate from total population or NHIP enrollment or what ? clarify.

Reviewers' comments:

Reviewer's Responses to Questions

**Comments to the Author**

1. Is the manuscript technically sound, and do the data support the conclusions?

Reviewer #1: No

Reviewer #2: No

2. Has the statistical analysis been performed appropriately and rigorously?

Reviewer #1: No

Reviewer #2: No

3. Have the authors made all data underlying the findings in their manuscript fully available?

Reviewer #1: Yes

Reviewer #2: No

4. Is the manuscript presented in an intelligible fashion and written in standard English?

Reviewer #1: Yes

Reviewer #2: Yes

Reviewer #1: 1. Sample Size Justification

Comment: The Methods state that “A total of 468 insured patients … were included” but no a priori power or sample-size calculation is provided to justify this number for the planned multi-ordinal logistic regressions (pp. 13–14, L126–131).

Recommendation: Please include a power analysis—specifying expected effect sizes, number of predictors, desired power (e.g., 80 %), and α-level—to confirm that n = 468 is sufficient to detect meaningful associations.

2. Handling of Likert-Scale Data

Comment: Satisfaction domain scores are calculated by averaging 18 PSQ-III items on a 1–5 Likert scale (pp. 20–21, L223–228). Treating ordinal data as continuous may bias estimates if scale assumptions are violated.

Recommendation: Report Cronbach’s α for each domain in the full sample (not only the pre-test) to confirm internal consistency.

Conduct sensitivity analyses using ordinal methods (e.g., factor analysis on the ordinal items or GEE for ordinal outcomes) to verify that mean-based scores yield similar results.

3. Choice of Multi-Ordinal Logistic Regression

Comment: The manuscript uses multi-ordinal logistic regression for three outcome categories (“dissatisfied,” “neutral,” “satisfied”) but does not report testing the proportional-odds assumption (pp. 8–9, L37–42).

Recommendation: Provide diagnostics (e.g., Brant test) for the proportional-odds assumption. If violated, consider partial proportional-odds or multinomial logistic models to avoid biased odds ratios.

4. Model Selection and Overfitting

Comment: Variable selection used stepwise forward selection based on AIC (pp. 15–16, L25–28). Stepwise methods can inflate Type I error and risk overfitting when many predictors are considered.

Recommendation:

Report the number of parameters relative to events per parameter in each domain model.

Consider penalized approaches (e.g., LASSO) or cross-validation to assess model stability.

Present full (“all-covariate”) models alongside the reduced models to demonstrate robustness of selected predictors.

5. Presentation of Regression Results

Comment: Table 4 displays only statistically significant predictors, obscuring potentially important confounders (pp. 24–25, L249–252).

Recommendation: Publish complete regression tables—including non-significant covariates—with AORs, 95 % CIs, and p-values for all variables to enhance transparency and allow readers to assess confounding.

6. Adjustment for Multiple Comparisons

Comment: Conducting seven separate domain-specific regressions increases the risk of spurious findings (pp. 24–25, Table 4).

Recommendation: Apply a multiple-testing correction (e.g., Bonferroni or Benjamini–Hochberg) or explicitly state that analyses are exploratory, to contextualize p-values—particularly those near α = 0.05.

Reviewer #2: Dear Authors,

We greatly appreciate the effort in your study assessing patient satisfaction under Nepal’s NHIP. To further enhance the clarity and rigor of your manuscript, we suggest the following statistical and methodological clarifications and improvements:

1. Justification of Reference Categories in Regression Models

In your ordinal logistic regression models, the choice of reference groups (e.g., "no formal education" for education level, "age below 20" for age, "agriculture" for occupation) appears arbitrary and lacks a clear rationale. Reference categories should ideally be selected based on:

- Policy relevance (e.g., the majority or socially normative group),

- Interpretation convenience (e.g., stable or meaningful baseline),

- Statistical stability (e.g., groups with sufficient sample size).

Using the first-listed category by default can lead to unstable or inflated odds ratios, particularly when categories differ significantly in sample size or variance (e.g., age below 20 vs. above 60 years in age groups). We recommend re-examining the choice of reference groups and explicitly justifying them within the methods section.

2. Incomplete Reporting of Multinomial Logistic Regression Results

In Table 4, the regression results are presented selectively, often showing odds ratios only for one or two categories (e.g., only "60+ age group" without comparisons to other age groups). This selective presentation limits the reader's ability to:

- Compare all levels within a categorical variable,

- Understand the overall effect pattern of the variable,

- Detect potential suppression or confounding effects.

If space is limited, we recommend including full regression tables (with all categories) as supplementary material, and clearly stating whether the table includes only statistically significant.

3. Lack of linkage between descriptive and inferential statistics

Table 1 provides useful descriptive statistics on patients’ socioeconomic characteristics, but lacks inferential statistical analysis to assess whether the categorizations used are appropriate and whether there are significant differences across groups. When variables have three or more categories, statistical tests such as ANOVA or chi-square tests can be employed to compare satisfaction levels across categories. These analyses would help clarify the structure of associations between patient characteristics and satisfaction, and reinforce the rationale behind the regression model specifications.

4. Disconnection Between Analytical Findings and Conclusions

The main findings of your regression models emphasize associations between patient satisfaction and socioeconomic characteristics (e.g., age, education, chronic illness). However, the conclusions emphasize only domain-level satisfaction (e.g., low satisfaction in financial and accessibility aspects) and call for broader health system strengthening—without referencing the regression findings. This weakens the internal logic of the study.

We suggest aligning the conclusions more directly with your analytical results—for instance, noting that vulnerable groups (e.g., older adults, lower-educated, subsidized patients) were systematically less satisfied, and that targeted policy actions may be needed for these groups.

**Do you want your identity to be public for this peer review?** For information about this choice, including consent withdrawal, please see our Privacy Policy

Reviewer #1: No

Reviewer #2: No

---

## [Author Response · Author response to Decision Letter 1]

23 Aug 2025

Response to Reviewer’s Comments

We would like to thank all the reviewers for their time and effort to read and comment on our paper. Their comments are constructive and the paper has benefitted from it. Below we present the point-by-point response to the reviewer’s comments.

REVIEWER 1

1. Sample Size Justification

Comment: The Methods state that “A total of 468 insured patients … were included” but no a priori power or sample-size calculation is provided to justify this number for the planned multi-ordinal logistic regressions (pp. 13–14, L126–131).

Recommendation: Please include a power analysis—specifying expected effect sizes, number of predictors, desired power (e.g., 80 %), and α-level—to confirm that n = 468 is sufficient to detect meaningful associations.

Response: Thank you for the comment.

We have added the following statement in the sampling section:

We calculated the sample size using the prevalence formula: n= Z2(p(1-p))/d2keeping prevalence at 50%. This formula gave us the minimum sample size of 385. With addition of a 20% non-response rate, the sample was calculated as: 462.

The data was collected from a total sample of 468 at the end of data collection period.

2. Handling of Likert-Scale Data

Comment: Satisfaction domain scores are calculated by averaging 18 PSQ-III items on a 1–5 Likert scale (pp. 20–21, L223–228). Treating ordinal data as continuous may bias estimates if scale assumptions are violated.

Recommendation: Report Cronbach’s α for each domain in the full sample (not only the pre-test) to confirm internal consistency.

Conduct sensitivity analyses using ordinal methods (e.g., factor analysis on the ordinal items or GEE for ordinal outcomes) to verify that mean-based scores yield similar results.

Response: Thank you for this valuable suggestion.

Cronbach's alpha was calculated as 0.8037. We have reported it in the method section as well.

We would like to clarify that our analysis already treats each of the 7 domains as ordinal outcomes, following the standard structure and scoring guidance of the PSQ-III tool (https://www.rand.org/health-care/surveys_tools/psq.html#:~:text=The%20PSQ%2DIII%20is%20a,III%20is%20also%20available%20below.)

In our analysis, we had used ordinal logistic regression (ologit) to model the relationship between each domain and relevant predictors. In the revised version, we have presented our findings by employing the Generalized ordered logistic regression.

The methods section has been revised accordingly.

3. Choice of Multi-Ordinal Logistic Regression

Comment: The manuscript uses multi-ordinal logistic regression for three outcome categories (“dissatisfied,” “neutral,” “satisfied”) but does not report testing the proportional-odds assumption (pp. 8–9, L37–42).

Recommendation: Provide diagnostics (e.g., Brant test) for the proportional-odds assumption. If violated, consider partial proportional-odds or multinomial logistic models to avoid biased odds ratios.

Response: We acknowledge your important point regarding the proportional odds assumption in ordinal logistic regression.

We have employed the Generalized Ordered Logistic Regression (gologit2) with the autofit option, for the partial proportional odds.

The results of this model are now reported in the revised manuscript (Table 4).

4. Model Selection and Overfitting

Comment: Variable selection used stepwise forward selection based on AIC (pp. 15–16, L25–28). Stepwise methods can inflate Type I error and risk overfitting when many predictors are considered.

Recommendation: Report the number of parameters relative to events per parameter in each domain model.

Consider penalized approaches (e.g., LASSO) or cross-validation to assess model stability.

Present full (“all-covariate”) models alongside the reduced models to demonstrate robustness of selected predictors.

Response: Thank you for this important comment. We recognize the limitations of stepwise selection methods, including the potential for overfitting and inflated Type I error, particularly when numerous predictors are evaluated.

We have revised our modality. We selected the best-fit model using stepwise selection method. Following this, we applied the Generalized Ordered logistic Regression with the autofit option to account for the partial proportional odds assumption. We have revised the analysis section likewise.

We have presented the reduced model in Table 4 and also presented the full model as supplemental file.

5. Presentation of Regression Results

Comment: Table 4 displays only statistically significant predictors, obscuring potentially important confounders (pp. 24–25, L249–252).

Recommendation: Publish complete regression tables—including non-significant covariates—with AORs, 95 % CIs, and p-values for all variables to enhance transparency and allow readers to assess confounding.

Response: Thank you for your feedback.

We have reported the full model table as Supplemental File.

6. Adjustment for Multiple Comparisons

Comment: Conducting seven separate domain-specific regressions increases the risk of spurious findings (pp. 24–25, Table 4).

Recommendation: Apply a multiple-testing correction (e.g., Bonferroni or Benjamini–Hochberg) or explicitly state that analyses are exploratory, to contextualize p-values—particularly those near α = 0.05.

Response: We have reported the Bonferroni corrected P-value in Table 4.

Also, we have stated that the analysis is exploratory. (In method section)

Reviewer 2

We greatly appreciate the effort in your study assessing patient satisfaction under Nepal’s NHIP. To further enhance the clarity and rigor of your manuscript, we suggest the following statistical and methodological clarifications and improvements:

1. Justification of Reference Categories in Regression Models

In your ordinal logistic regression models, the choice of reference groups (e.g., "no formal education" for education level, "age below 20" for age, "agriculture" for occupation) appears arbitrary and lacks a clear rationale. Reference categories should ideally be selected based on:

- Policy relevance (e.g., the majority or socially normative group),

- Interpretation convenience (e.g., stable or meaningful baseline),

- Statistical stability (e.g., groups with sufficient sample size).

Using the first-listed category by default can lead to unstable or inflated odds ratios, particularly when categories differ significantly in sample size or variance (e.g., age below 20 vs. above 60 years in age groups). We recommend re-examining the choice of reference groups and explicitly justifying them within the methods section.

Response: Thank you for this insightful observation. We appreciate the importance of selecting reference categories thoughtfully to ensure both interpretability and statistical stability.

In the revised version, we have taken the reference groups based on the information from this article (https://dhsprogram.com/pubs/pdf/WP199/WP199.pdf).

We have referred to the groups with highest enrollment rates as reference groups.

2. Incomplete Reporting of Multinomial Logistic Regression Results

In Table 4, the regression results are presented selectively, often showing odds ratios only for one or two categories (e.g., only "60+ age group" without comparisons to other age groups). This selective presentation limits the reader's ability to:

- Compare all levels within a categorical variable,

- Understand the overall effect pattern of the variable,

- Detect potential suppression or confounding effects.

If space is limited, we recommend including full regression tables (with all categories) as supplementary material, and clearly stating whether the table includes only statistically significant.

Response: Thank you for your feedback.

We have presented the results for all categories. Also, the full table has been uploaded as the supplemental file 4.

3. Lack of linkage between descriptive and inferential statistics

Table 1 provides useful descriptive statistics on patients’ socioeconomic characteristics, but lacks inferential statistical analysis to assess whether the categorizations used are appropriate and whether there are significant differences across groups. When variables have three or more categories, statistical tests such as ANOVA or chi-square tests can be employed to compare satisfaction levels across categories. These analyses would help clarify the structure of associations between patient characteristics and satisfaction, and reinforce the rationale behind the regression model specifications.

Response: The chi-square values of the variables for each domain are added as supplemental file. However, we did not rely on the chi-square values for model selection.

Methodologically, we conducted a stepwise selection method for choosing the best-fit model, followed by the Generalized ordered logistic regression with partial proportional odds model.

4. Disconnection Between Analytical Findings and Conclusions

The main findings of your regression models emphasize associations between patient satisfaction and socioeconomic characteristics (e.g., age, education, chronic illness). However, the conclusions emphasize only domain-level satisfaction (e.g., low satisfaction in financial and accessibility aspects) and call for broader health system strengthening—without referencing the regression findings. This weakens the internal logic of the study.

We suggest aligning the conclusions more directly with your analytical results—for instance, noting that vulnerable groups (e.g., older adults, lower-educated, subsidized patients) were systematically less satisfied, and that targeted policy actions may be needed for these groups.

Response: Thank you for your feedback.

We have now revised the conclusion with addition of information from the associated variables.

---

## [Decision Letter · Decision Letter 1]

13 Sep 2025

Dear Dr. Marasini,

Thank you for submitting your manuscript to PLOS ONE. After careful consideration, we feel that it has merit but does not fully meet PLOS ONE’s publication criteria as it currently stands. Therefore, we invite you to submit a revised version of the manuscript that addresses the points raised during the review process.

Please address the comments of the reviewer and also check the grammatical/spell errors throughout the text as well as rationale for the reference value in methods/data analysis part. 

We look forward to receiving your revised manuscript.

Kind regards,

Kshitij Karki, MPH, MA

Academic Editor

PLOS ONE

Journal Requirements:

Additional Editor Comments:

Thank you for the revision. Please go through the minor comments from the reviewer. Also, I would like to request you to go through the grammatical/spell errors in the manuscripts (as in Line 123).

You can also write in methods/ data analysis part - why you choose the reference value for bivariate/multivariate analysis.

Reviewers' comments:

Reviewer's Responses to Questions

**Comments to the Author**

Reviewer #1: All comments have been addressed

Reviewer #2: All comments have been addressed

2. Is the manuscript technically sound, and do the data support the conclusions?

Reviewer #1: Yes

Reviewer #2: Yes

3. Has the statistical analysis been performed appropriately and rigorously?

Reviewer #1: Yes

Reviewer #2: Yes

4. Have the authors made all data underlying the findings in their manuscript fully available?

Reviewer #1: Yes

Reviewer #2: Yes

5. Is the manuscript presented in an intelligible fashion and written in standard English?

Reviewer #1: Yes

Reviewer #2: Yes

Reviewer #1: (No Response)

Reviewer #2: Thank you for carefully revising your manuscript in response to the previous review comments. I appreciate the effort you have made to incorporate the suggested changes, and I believe the paper has improved as a result. I would like to offer a few additional minor comments, which I believe, if addressed, will further strengthen your work and contribute to making it an even better paper.

- Please provide additional details on the generalized ordered logistic regression, including how the proportional odds assumption was tested and whether it was satisfied.

- In Table 1, please add a note explaining the presence of missing data for variables where N is less than 468.

**Do you want your identity to be public for this peer review?** For information about this choice, including consent withdrawal, please see our Privacy Policy

Reviewer #1: No

Reviewer #2: No

---

## [Author Response · Author response to Decision Letter 2]

17 Sep 2025

Editor Comments

Please go through the minor comments from the reviewer. Also, I would like to request you to go through the grammatical/spell errors in the manuscripts (as in Line 123).

Response: We have carefully reviewed the entire manuscript and grammatical as well as spelling errors have been corrected to improve clarity and readability.

You can also write in methods/ data analysis part - why you choose the reference value for bivariate/multivariate analysis.

Response: Thank you for the comment. For the analysis, we conducted stepwise selection method for selecting the best-fit model in our analysis. This model was then assessed for partial proportional odds assumption using generalized ordered logistic regression.

We have included these details in our methods section as well.

Reviewer’s comments

Please provide additional details on the generalized ordered logistic regression, including how the proportional odds assumption was tested and whether it was satisfied.

Response: Thank you for the comments. We have added the following statement in methods section for clarity to the readers.

We assessed the proportional odds assumption for each variable included in the generalized ordered logistic regression. For variables that satisfied the proportional odds assumption, the threshold outcomes are presented as “All.” For variables that did not satisfy the assumption, separate adjusted odds ratios (AORs) are reported for each comparison (i.e., satisfied vs. dissatisfied and satisfied vs. neutral).

In Table 1, please add a note explaining the presence of missing data for variables where N is less than 468.

Response: Thank you for the comment. We have added a note in the footnote of the Table.

---

## [Decision Letter · Decision Letter 2]

26 Sep 2025

Patient Satisfaction with Healthcare Services among Health Insurance Program Beneficiaries in Nepal: A Cross-Sectional Study

PONE-D-25-31570R2

Dear Dr. Sabina Marasini,

We’re pleased to inform you that your manuscript has been judged scientifically suitable for publication and will be formally accepted for publication once it meets all outstanding technical requirements.

Kind regards,

Kshitij Karki, MPH, MA

Academic Editor

PLOS ONE

Additional Editor Comments (optional):

Reviewers' comments:

Reviewer's Responses to Questions

**Comments to the Author**

Reviewer #2: All comments have been addressed

2. Is the manuscript technically sound, and do the data support the conclusions?

Reviewer #2: Yes

3. Has the statistical analysis been performed appropriately and rigorously?

Reviewer #2: Yes

4. Have the authors made all data underlying the findings in their manuscript fully available?

Reviewer #2: Yes

5. Is the manuscript presented in an intelligible fashion and written in standard English?

Reviewer #2: Yes

Reviewer #2: (No Response)

**Do you want your identity to be public for this peer review?** For information about this choice, including consent withdrawal, please see our Privacy Policy

Reviewer #2: No

---

## [Editor Report · Acceptance letter]

PONE-D-25-31570R2

PLOS ONE

Dear Dr. Marasini,

I'm pleased to inform you that your manuscript has been deemed suitable for publication in PLOS ONE. Congratulations! Your manuscript is now being handed over to our production team.

Kind regards,

on behalf of

Dr. Kshitij Karki

Academic Editor

PLOS ONE